# Plant-Based Nano-Emulsions as Edible Coatings in the Extension of Fruits and Vegetables Shelf Life: A Patent Review

**DOI:** 10.3390/foods12132535

**Published:** 2023-06-29

**Authors:** Vanja Travičić, Teodora Cvanić, Gordana Ćetković

**Affiliations:** Faculty of Technology Novi Sad, University of Novi Sad, Bulevar cara Lazara 1, 21000 Novi Sad, Serbia; teodora.cvanic@uns.ac.rs (T.C.); gcetkovic@uns.ac.rs (G.Ć.)

**Keywords:** edible coatings, nano-emulsions, fruit and vegetable shelf life, innovations, patent review

## Abstract

Fresh fruits and vegetables are important sources of minerals, vitamins, fibers, and antioxidants, essential for human well-being. However, some fruits and vegetables are highly perishable with a very short shelf life during storage. Serious consumer concern over the use of chemical preservatives for this purpose has led to a green revolution and a sustainable era where the design and fabrication of edible coatings have attracted considerable interest. In recent years, scientific communities have paid great attention to the development of bio-based edible coatings to extend the postharvest shelf life of fruits and vegetables. Furthermore, nanotechnology has been distinguished as a great strategy for improving coating properties, including a better water barrier and better mechanical, optical, and microstructural properties, as well as gradual and controlled release of bioactive compounds. In this work, patent articles on plant-based nano-emulsions as edible coatings in the extension of fruit and vegetable shelf life were reviewed. The Patentscope search service and Espacenet portal were used, applying a query strategy composed of mesh terms and inclusion criteria. Through database searching, a total of 16 patent documents met the inclusion criteria. Further, to demonstrate the innovation trends in this topic, all relevant patents are described at the end of the study, along with the components, technology, application, and advantages of developed preparations.

## 1. Introduction

The COVID-19 pandemic has affected consumer behaviors in all areas of life. These rapid shifts have important implications for retailers and consumer packaged goods companies. Many of the longer-term changes in consumer behavior are still forming, giving companies an opportunity to link with science, generate innovative ideas, and find solutions to satisfy human needs. Namely, the COVID-19 pandemic has led to surging demand for products that support the overall maintenance of health and wellness, urging consumers to improve their nutritional intake and consume safer and healthier food. This has resulted in an increased demand for fresh and organic food, including fruits and vegetables [1].

Fresh fruits and vegetables are perishable commodities that require technologies to extend their postharvest quality and shelf life. It is well known that after harvest, fruits and vegetables continue the respiration process, consuming O_2_ and releasing CO_2_, heat, and water. Furthermore, components such as lipids, proteins, organic acids, and carbohydrates are metabolized even after the separation from the mother plant [2]. According to the Food and Agriculture Organization (FAO), more than 1.3 billion tons of food are squandered annually; fruits and vegetables contributed 40–50% of the total generated waste. Many traditional food preservation techniques are being employed to preserve food; however, limited methods are available to extend the shelf life of fruits and vegetables. For this purpose, a highly familiar method is the use of thin-layered edible coatings or films [3].

The application of edible coatings or films for the extension of fruit and vegetable shelf life dates to the 12th century in China, when citrus fruits (oranges and lemons) were coated with wax to preserve their quality [4]. This method was quite effective for its time, so the wax layer on fruits was used for centuries in lack of more efficient methods. Nowadays, edible coatings can be prepared using polysaccharides, proteins, lipids, or a composition of these compounds. These edible coatings act as a barrier against the various atmospheric gases, humidity or water vapors, oxygen, carbon dioxide, and microbes, thus helping to decrease the respiration and oxidation reaction rates in fruits and vegetables. However, none of the three constituents of edible coatings can provide the needed protection by themselves; usually, the active agents are added to edible coatings to improve the coated product’s antimicrobial, antioxidant, flavor, color, or nutritional properties [5]. Regarding active agents, the use of essential oils and other plant extracts has been evaluated in the development of edible coatings, given their demonstrated potential to inhibit microbial growth, which damages products and decreases shelf life. These coatings can also utilize the antioxidant capacity of plant extracts and essential oils to scavenge free radicals, decrease deterioration rates, and prevent oxidation reactions [6]. Today, nanotechnology represents an area of opportunity for developing vehicles to transport certain active agents, such as polyphenols and carotenoids, with antimicrobial and antioxidant properties. The evolution of nanotechnology in the food and agriculture sector has shown tremendous opportunities to make food preservation and packaging possible without causing any side effects on health and the environment [7].

In 2021, de Oliveira Filho et al. [7] reported the distribution of publications related to “nanoemulsions as edible coating for fruits and vegetables”, for the period from 2005 to 2021; the number of studies on the topic has increased considerably over the past few years, demonstrating the scientific community’s increased interest in the topic. Moreover, several papers have reviewed the recent advances in the development of edible coating technologies with antimicrobial activity to extend the shelf life of fruits that use various plant materials and nanoscale materials [3,4,5,6,7,8]. However, this paper aims to review the state-of-the-art patents on edible coatings in the extension of fruit and vegetable shelf life [8,9]. The first patent on the use of edible coatings dates to 1916; the patented method was created by Hofman, which is related to preserving whole fruits with molten wax [10]. After that, in 1972, Bryan patented a method to preserve grapefruit halves with an edible coating based on low methoxyl pectin and locust bean gum dispersed in grapefruit juice [11]. Generally, the scope of this review is defined by inclusion criteria (plant-based product in a nano-emulsion form). We also include a discussion of the most relevant features available in the patent document (components, technology, application).

## 2. Green Revolution against Chemical Pesticides

The fast decline in the quality attributes of fresh fruits and vegetables during the postharvest period is a problematic issue on a global level. For this purpose, for many years farmers have been using chemical preservatives such as chlorine dioxide, nitric oxide, salicylic acid, and 1-methyl cyclopropane to increase the shelf life and maintain the postharvest quality of fruits and vegetables [12]. It is well known that such preservatives bring potentially dangerous compounds into the food chain and cause harmful effects on consumers’ health and biodiversity. According to the World Health Organization (WHO), about 3 million annual cases of chemical pesticide poisoning and up to 220,000 deaths are recorded. The human body’s exposure to pesticides can cause DNA damage and disorders such as diabetes mellitus, Parkinson’s disease, and Alzheimer’s disease. The general problem is that in most developed countries, pesticide residues on fruits and vegetables have been tracked for decades, while those in developing countries are not properly recorded. For this reason, consumers are currently concerned about the use of these kinds of sprayed fruits [12].

As the answer to consumer demands, the green revolution was started, meaning the partial or complete removal of chemically synthesized preservatives from food stuffs, which are used to extend food shelf life. Therefore, the investigation of natural preservatives obtained from various plant materials is of great importance, as well as their possible application. They play a crucial role in edible packaging or active packaging due to their antioxidant, antibacterial, and anti-fungal activities [13]. Besides active compounds, synthetic material for food packaging can be replaced by natural, degradable, and safe ingredients in the form of edible films and coatings. Edible films and coatings represent a revolution in food packaging because they are regarded as safe and eco-friendly; moreover, they can decrease the rate of respiration and ripening, decrease ethylene, control moisture loss, and eliminate microbial activities [14]. The trend of avoiding chemically synthesized preservatives cannot negatively impact food safety as it is crucial for sustaining life, promoting good health and economic growth. Therefore, it is of immense importance to prevent and provide excellent food safety while reducing food waste [15]. However, constant improvement in food protection is necessary to track the progress of society, as the population is expected to increase to 10.9 billion people by 2050 [16].

## 3. Patents Search Strategy

First, a preliminary search was carried out for patent review articles on plant-based nano-emulsions as edible coatings in the extension of fruit and vegetable shelf life. Mesh terms used for the search were “Review aritcle” AND “Patent” OR “Patents”, AND “Nanoemulsion” AND “Edible Coatings” OR “Edible Films” AND “Fruits” OR “Vegetables” AND “Shelf-life”; zero results were obtained from the SCOPUS database, which implies that this paper will be helpful for the science community and all those interested in this topic.

The patent search was performed on 4 May 2023, and the main strategy was based on a scooping study to find out how many patents had been registered related to plant-based nano-emulsions as edible coatings with a focus on the extension of fruit and vegetable shelf life. For this purpose, two platforms were utilized. Namely, the World Intellectual Property Organization (WIPO) and the European Patent Office (EPO) allowed us to search their registered patents using free databases, including Patentscope search service and Espacenet. This process involved a query strategy composed of keywords and inclusion criteria, as shown in Figure 1. The search was filtered to include only the patent documents for the period from January 2012 to April 2023. Documents that were not related to the topic were disregarded after reading the abstract, and documents that were not available in English were disregarded as well.

## 4. Analysis of Documents and Selection of Relevant Patents

During a search of patents from 2012 to 2023, 833 patent documents were identified through database searching (Espacenet = 362 and Patentscope = 471). A total of 16 patents met the inclusion criteria, and eligible patents are presented in Table 1.

Figure 2 shows the publication dates of patents (a) and the distribution of found patents by country (b), as well as the types of affiliation and partnerships involved in the patents that concern plant-based nano-emulsions as edible coatings in the extension of fruit and vegetable shelf life. In general, the granted patents were recorded in the last five years; until 2019, not a single patent was registered. The documents related to this topic were mostly registered in China (14 patents); one patent each is registered in the United States and Singapore. The profile observed in the distribution of patent holders emphasizes that this topic is still under development in the scientific community, since 12 of these patents were granted to universities and institutes/research centers.

### Review Based on Relevant Patents

In 2019, Li et al. [22] invented a food coating film, which particularly relates to an antibacterial emulsion with pure essential oil. The invention focuses on the preparation method and application thereof. Briefly, the aqueous phase of the emulsion was a mixture of chitosan and aqueous glacial acetic acid; the aqueous phase had a pH of 6.5, and the chitosan concentration was 0.1% to 0.3%. Further, the natural flavor aldehyde was any one of the following: cinnamaldehyde, citral, and citronellal. The plant essential oil was any one of the following: clove oil, rose essential oil, chamomile essential oil, bitter orange essential oil, lavender essential oil, and thyme essential oil. The surfactant was a food-grade nonionic surfactant, monocaprylin or lauric acid monoglyceride or monoglyceride or polyglycerin fatty acid ester. The chitosan had a viscosity average molecular weight of 10,000 to 280,000 and a degree of deacetylation of 80 to 95%. In addition to the detailed procedure for the preparation method of this emulsion, the inventors reported in their patent application the results of five specific embodiments for better illustration of the invention. In summary, the advantages and positive effects of their invention are: (I) the antibacterial activity of the prepared emulsion to *Escherichia coli* and *Staphylococcus aureus* is superior to that of single-plant essential oil; (II) the content of the surfactant is low; (III) the emulsion does not contain edible fat and oil; (IV) the preparation conditions are mild, and the emulsion is safer to use and stable, has a long storage period, is used after being diluted, and can be widely applied to rot prevention and the fresh preservation of fresh food.

Dan et al. [27] revealed a method for the production and utilization of cinnamon essential oil Pickering emulsion by different polymer solutions. They suggested using 45–55% of compounded oil phases where the main component was a cinnamon essential oil, while suggestions for polymer solutions were 18–23% of a zein solution, 7–12% of a pectin solution, and 17–23% of a xanthan gum solution. In addition to cinnamon essential oil, in the compounded oil phase was a lemon essential oil, in the volume ratio of 5:1, and coconut oil, where the volume ratio of cinnamon oil/coconut oil was 1:1~1:20. The source of proteins can be soy protein, modified natural starch, cellulose, and derivatives thereof, while pectin can be chosen from hawthorn pectin, apple pectin, and sugar beet pectin. In this work, the following procedure was described: (1) obtain zein solution by dissolving it in 80% ethanol and preparing an aquatic solution under an ultrasonic process; (2) obtain pectin and xanthan gum solution by dissolving them in ultrapure water at 40 °C and 25 °C, respectively; (3) obtain composite nanoparticle solution dropwise via zein aquatic solution in pectin solution under magnetic stirring at 200 rpm; (4) add the compound oil phase to the nanoparticle solution prepared in step (3) under high-speed dispersion conditions (speed: 12,000~16,000 rpm; time: 2~4 min), followed by adding the xanthan gum solution to obtain the final product, which has an oil phase droplets size of 20 to 50 μm. Cinnamon-essential-oil-loaded Pickering emulsion can be used for the preservation of all kinds of fruits and vegetables in the postharvest treatments and for the extension of shelf life while expressing a significant antibacterial effect and a broad spectrum of inhibition.

The invention of antibacterial liquid preservatives was explained in the work of Wang and Xue [30], who designed a lacquer wax composite nanometer silver liquid with ginkgo leaf lipid compound that possesses a strong antibacterial effect against *Staphylococcus aureus*, *Bacillus subtilis*, and other fungi. Several steps were conducted in order to prepare it: (1) wax nano-silver emulsion, (2) ginkgo biloba concentrate, (3) ginkgo lipid nano silver emulsion, (4) shellac emulsion, and (5) wax composite preservative. The first step included the following compounds in the following mass ratios: refined wax (1:10), emulsifier (5:20), nano-silver liquid (0.1:3), and deionized water (20:90). The second step was subcritical extraction of the ginkgo biloba leaves. The third step included stirring ginkgo biloba fat concentrate (1:100), chitosan (1:5), nano-silver liquid (0.1:5), and deionized water (85:95) in the stated mass ratios. For the preparation of shellac, emulsion was mixed with mass percentages of 5~15% shellac, 1~5% morpholine, and 1~2% propylene glycol until it was melted, and after that 1~4% oleic acid, 1~2% ammonia water, and 75~95% were added with deionized water. In the fifth step, the nano-silver emulsion, the ginkgo lipid nano-silver emulsion, and the shellac emulsion were stirred in mass percentages of 15 to 25, 20 to 40, and 50 to 70, respectively. The obtained emulsion had a particle size of 400–1000 nm, improved film-forming properties, and good permeability and water repellency while possessing preservative characteristics verified by a reduction of 30% to 60% of the citrus decay rate in comparison with the control.

Another application from 2019 was proposed by Wang et al. [23], in their invention is described an essential oil nano-emulsion emulsified by xanthan gum. The invented formulation contained the following volume percentages: 1 to 20 percent of composite oil phase where clove essential oil was a main component; 60 to 78 percent of a xanthan gum solution; and 15 to 23 percent of a starch octenyl succinate solution. The oil composition phase comprised a volume of 85% of clove essential oil, 9% thyme essential oil, and 6% lemon essential oil, while concentrations of xanthan gum and starch octenyl succinate were 0.08 to 0.6% and 8 to 12%, respectively. Separately, a preparation of the main ingredients was made via stirring, while mixing of the composite oil phase with the continuous phase (xanthan gum and starch octenyl succinate solution) was performed by a high-speed disperser to obtain a uniform and stable nano-emulsion. The nano-emulsion described had several advantages such as (I) a small oil particle size and high dispersion degree; (II) no droplet accumulation and demulsification; (III) high storage stability, 20 days at 25 °C; (IV) a simple and convenient preparation process; (V) a fast, precise, mild, and effective effect on the preservation of fresh-cut fruits and vegetables.

The fresh-keeping method for fruits and vegetables invented by Zou et al. [24] was a food-grade, nontoxic, and harmless method for effectively solving the problem of spoilage of fruits and vegetables. The invented product was a double emulsion that consists of the following parts: (1) internal water phase, where cinnamon essential oil, gluconolactone, and sodium alginate were dissolved in distilled water, and mass concentrations were 0.75–1.5%, 3.5–5.0%, and 1–1.5%, respectively; (2) oil phase, where salicylic acid and polyglycerol ricinoleate were added to corn oil and stirred, and mass concentrations were 0.15–0.25% and 5–8%, respectively; and (3) external water phase, where Tween and sodium alginate was added to distilled water and mixed, and then calcium carbonate was added and stirred and ultrasonically dispersed, where mass concentrations were 5–10%, 1.5–3%, and 0.4–1.0%, respectively. The production of a double emulsion was performed in two steps; firstly, a water-in-oil emulsion was prepared from the internal water phase and the oil phase by using shear dispersion and microjet treatment, following the mass ratio 4:6–3:7; secondly, a double emulsion was obtained by shearing and dispersing the external water phase and the water-in-oil emulsion, following the mass ratio 6:4–7:3. The suggested implementation of the invented method was the use of a previously described double nano-emulsion via spraying followed by bagging the fruits and vegetables in order to allow the essential oils of cinnamon and salicylic acid to express their full preservation effect on the fruits and vegetables.

Trinetta and Yucel [17] invented a lipid nano-emulsion-doped antimicrobial packaging film. They aimed to promote the film as an effective active packing technology, which included cooling as a triggering mechanism of antimicrobials’ release, to control postharvest disease and extend the shelf life of plants. This film generally comprised an antimicrobial agent, such as an essential oil or essential oil mixtures, encapsulated in a carrier and dispersed in a film matrix. The film matrix comprised a polysaccharide, such as pullulan, and can be used in packaging systems. The authors highlighted that films may comprise only edible components, generally recognized as safe (GRAS), and/or approved by FDA. The films allow the slow release of antimicrobial compounds over time, thereby controlling and/or inhibiting postharvest disease and extending the shelf life of plants, including vegetables and/or small fruit specialty crops.

Malanati Ramos et al. [20] revealed the method for producing a nano-emulsion comprising nano-encapsulated natural antioxidants, disclosing the specific formulation and the application thereof for the preservation of different natural food products or derivatives such as fresh or minimally processed fruit, vegetables, cereals, and juices. The invented formulation comprised four main steps: (1) isolation of natural antioxidants from fruit, vegetable, and cereal waste; (2) encapsulation of the natural antioxidants; (3) formation of the nano-emulsion with natural antioxidants; (4) freeze-drying of the nano-emulsion formed. In a general sense, their intention was to design a method for producing nano-emulsions with a high antioxidant power from fruit and/or vegetable and/or cereal waste that was efficiently encapsulated, easy to process, and without organic chemical additives in the final product. The aim of this nano-emulsion was to preserve and/or enhance the nutritional and organoleptic properties of fresh and minimally processed products, for humans and animals, with food or nutraceutical grade. The key aspect of the invention was a thin, nanometer-sized layer on the food, which prevented gas and fluid exchange with the environment, enhanced with selected antioxidants, the role of which was like an enzyme that slows down or inhibits the biochemical decomposition and oxidation reactions of the food. This provided fresh and minimally processed foods with a longer shelf life and meant that the organoleptic quality of foods to be frozen improved upon thawing.

Zhang et al. [19] developed modified atmosphere packaging method for fresh-cut fruits and vegetables by combining a nano-coating film with a microporous film. According to this method, the fresh-cut fruits and vegetables were preserved by adopting a lemon grass essential oil nano-emulsion coating and laser micro-pore film synergistic treatment technology. This invention could effectively control the total number of microorganisms and the total number of mold yeasts on the surfaces of fresh-cut fruits and vegetables and reduce the respiration rate of products. Furthermore, the gas concentration and the relative humidity in the package were effectively controlled, the water loss and the quality reduction of the product were delayed, the edible safety of the product was ensured, and the shelf life of the product was prolonged.

An invention which complies with halal, kosher, and vegan dietary requirements and relates to a film-forming composition for extending the shelf life of food products was disclosed by Antipina and Belova [21]. The inventors provided a method for producing a film-forming composition. Namely, the film-forming composition contained a hydrogel based on polysaccharides, and a water-soluble additive including a phenolic bioactive and an anti-fungal agent, wherein the hydrogel was formed as a matrix incorporated with the water-soluble additive. This invention was not limited to fruits and vegetables, and it possessed antimicrobial and anti-fungal properties. Beyond that, the films provided an aesthetic appearance to the coated commodities, protected them from UV light (premature over-ripening), and prevented evaporation of moisture (drying) caused by the process of transpiration that is characteristic of fruits and vegetables. Antimicrobial and anti-fungal activity of the films was based on a pathogen-suppressive mixture of low-cost food grade antimicrobial phenolic phytochemicals (e.g., tannic acid and gallic acid) and two anti-fungal food additives, potassium sorbate and sodium benzoate, incorporated into a film matrix made of plant polysaccharides (pectin or k-carrageenan). The synergistic effect of the phytochemicals and salts enabled the obtained films to inhibit both Gram-positive and Gram-negative bacteria and mold. The final formulation contained no ingredients of animal origin, and it is produced through a fully water-based method; thus, the coating films are compatible with vegan, halal, and kosher dietary requirements.

The following invention belongs to the technical field of the fresh keeping of vegetables and fruits and particularly relates to an application method of *Zingiber corallinum* Hance essential oil in the preparation of an edible spray for prolonging the shelf life of vegetables and fruits. More specifically, Zhao et al. [18] presented the following steps for the development of this product: enriching antioxidant components in *Zingiber corallinum* Hance essential oil by adopting a molecular distillation technology, preparing an essential oil nano-emulsion, and mixing the essential oil nano-emulsion with a film-forming solution prepared from chitosan and Arabic gum to prepare the edible spray. The inventors claimed that the raw materials of their product are edible and harmless to human bodies and the environment; moreover, antibacterial and antioxidant components are released in a slow-release mode, and the formed protective film can play a long-acting role in resisting bacteria and oxidation, so the shelf life of vegetables and fruits is further prolonged. They recommended spraying it evenly on the surface of freshly picked fruits and vegetables and waiting for the spray to dry naturally. This spray method can make the protective film attach to the surface of the fruits and vegetables in a thinner and more uniform way, while proving to be more convenient for use.

The invented method of Chen et al. [25] disclosed a preparation method of a *Rhizoma Zingiberis Recens* essential oil nano-emulsion. According to the authors, the preparation method comprised the following steps: independently weighing 1 part of the oil phase and 100 parts of the aqueous phases, adding a mixed liquid into a stirrer, stirring at the stirring speed of 800 turns/minute for 30 min, utilizing a high-speed shearing machine, regulating the revolving speed to 12,000 r/min, carrying out shearing for 5 min, and, finally, utilizing a high-pressure homogenizer to carry out homogenization six times under 100 MPa to obtain the stable essential oil nano-emulsion. Further, the authors presented the application of the prepared essential oil nano-emulsion in tomato preservation; they claimed that the *Rhizoma Zingiberis Recens* essential oil nano-emulsion had good stability and high antibacterial and anti-oxidization properties, so it could prolong the product shelf life of the tomato and was suitable for the freshness retention and corrosion prevention of fruit and vegetable products. The invention has the following advantages: (I) moderate viscosity, stable properties, and can be infinitely diluted with phosphate buffer; (II) strong stability and strong antibacterial properties. It can be used in combination with low temperature for the preservation and anticorrosion of fruit and vegetable products. Different from traditional chemical preservatives, it is a natural antibacterial agent to ensure food safety; (III) the ginger essential oil nano-emulsion has the effect of simultaneously inhibiting spoilage bacteria and pathogenic bacteria and at the same time inhibits the formation of the cell membrane of spoilage bacteria and pathogenic bacteria, so it can reduce the drug resistance of harmful microorganisms; (IV) with bovine lactoferrin as the sole surfactant, it is safer and reduces the cost.

In 2021, Xu et al. [32] presented their invention, which belongs to the field of food. Their invention particularly relates to a clove essential oil Pickering emulsion coating preservative, a preparation method, and the use thereof. The inventors suggested the following steps for the preparation: preparing 20 mg/mL of a zein solution from ethanol and zein powder, preparing 0.4–6 mg/mL of a sodium caseinate solution from ultrapure water and sodium caseinate, and magnetically stirring overnight; dropwise adding 20 mg/mL of the zein solution into 0.4–6 mg/mL of the sodium caseinate solution and magnetically stirring at 500 rpm for 30 min; evaporating the ethanol in the solution at 40 °C by using a rotary evaporator, wherein the evaporated ethanol was replaced by ultrapure water; adding clove essential oil and homogenizing at a high speed for 3 min to obtain a clove essential oil Pickering emulsion; and preparing a 1.5–2% *w*/*v* chitosan solution from 1% *v*/*v* acetic acid and chitosan, adding the clove essential oil Pickering emulsion into the chitosan solution, homogenizing at a high speed for 3 min, and carrying out ultrasonic treatment for 30 min to defoam, thereby obtaining the clove essential oil Pickering emulsion coating preservative. According to the patent description, the invented emulsion has a wide range of applications, especially in the field of the fresh keeping of fruits, vegetables, and meat products.

The following year, Feng et al. [26] invented a composite natural preservative prepared from green and environmentally friendly raw material components. The main advantages are a limited violation of active compounds due to incorporation in aquatic solution, easy storage and transportation, and simple preparation and manipulation. Moreover, this product is eco-friendly, healthy, nonhazardous, and nontoxic, and it is a green product with efficient preservation qualities. The main ingredients of this preservative product were tea saponin 1–4 mg/mL, tea tree oil nano-emulsion 3–5 mg/mL, and chitosan 10–20 mg/mL. Preparation of the tea tree oil nano-emulsion is essential due to its low water solubility and it being easily violated. The mass ratio of polysorbate-80 used as an emulsifier and tea tree oil is 2:3 for the production of the oil phase, while the water phase contained ethanol at a concentration of 30–50 g/L. The coarsed nano-emulsion was formed by stirring the oil and the water phase, which was followed by the use of ultrasonic homogenization, where ultrasonic power and time were 400–450 W and 10–20 min, respectively, to obtain the nano-emulsion. The final product was gained by mixing the previously described amounts of tea saponin, tea tree oil nano-emulsion, and chitosan. The proposed steps for use are to spray it evenly on the surface, let it dry for 15 to 30 min, and finally to pack the food in a polyethylene, fresh-keeping bag. The synergistic effect of the main constituents can provide efficient prolongation of the storage period of vegetables and ensure their appearance and quality.

Pan and Nie [31] developed a black pepper essential oil nano-emulsion, as well as a preparation method and application thereof. A black pepper essential oil nano-emulsion was prepared from the following raw materials in parts by weight: 3–6 parts of black pepper essential oil, 1.5–3.5 parts of surfactant, and 1–1.5 parts of co-surfactant, 85 to 95 parts of water. According to the authors, the proposed surfactant included one or more of the following: Tween 80, Span 80, soybean lecithin, APG1214, and sucrose fatty acid ester, while the preferred co-surfactant included one or more of the following: ethanol, glycerol, and n-butanol. In this invention, the effect of the fruit preservative containing black pepper essential oil nano-emulsion on mango postharvest anthracnose was investigated. The black pepper essential oil nano-emulsion disclosed by the invention has good antibacterial activity and can be prepared into a fruit preservative. This technology has many advantages, the source is wide, the cost is low, the use is convenient, and the use effect is excellent.

The invention of Qui et al. [29] disclosed a preparation method of a nano-structure lipid carrier based on medium- and long-chain diglyceride-embedded *Litsea cubeba* essential oil. In addition, the authors presented an application of the nano-structure lipid carrier in fruits and vegetables preservation. The development path consisted of the following steps: taking *Litsea cubeba* essential oil, vegetable oil, and molten medium and long-chain diglyceride as an oil phase; mixing the water phase into the oil phase, performing high-speed shearing and homogenizing, performing ultrasonication, and stirring and cooling in an ice bath to obtain the nano-structure lipid carrier based on the medium- and long-chain diglyceride-embedded *Litsea cubeba* essential oil. The preferred vegetable oil was at least one of the following: olive oil, soybean oil, corn oil, sunflower oil, tea seed oil, cottonseed oil, rice bran oil, peanut oil, or linseed oil; the emulsifier was one of the following: Tween 80, Tween 60, or Tween 20. This invention has several advantages: small particle size, high encapsulation efficiency, and good stability. Furthermore, this invention also solves the problem of the *Litsea cubeba* essential oil’s solubility; according to the authors, the *Litsea cubeba* essential oil is difficult to dissolve in water due to its oily nature and can be utilized due to its high volatility, resulting in efficient embedding and delayed release of the *Litsea cubeba* essential oil. The additional advantages of the final product are broad-spectrum of antibacterial activity and oxidation resistance; blueberry rot was effectively inhibited, the shelf life of blueberries was prolonged, and the economic value was improved.

A fruit nano-film preservative, invented by Sun et al. [28], was produced using curdlan octenyl succinate as the packaging material, while isobutyl paraben (5–10 parts) and ethylene oxide higher fatty alcohol were used as core materials. The fruit nano-coating preservative comprises the following raw materials in parts by weight: 5–10 parts of isobutyl paraben; 5–10 parts of ethylene oxide higher fatty alcohol; 10 to 20 parts of octenyl succinic acid curdlan gum ester; 0.2 to 1 part of surfactant; and 59 to 79.8 parts of water. Sucrose fatty acid ester was used as surfactant. The particle size of the nano-emulsion was about 200 nm. The inventors claim that the proposed product, in comparison with other products, not only it is coating the fruit more evenly, but the amount needed for use was also greatly reduced. Moreover, the inventors highlight the safety and reliability of consumption of this kind of preservative.

## 5. Future Perspectives and Research Needs

As aforementioned food waste is becoming a significant problem in modern society, the prolongation of food shelf life is becoming prioritized and receiving more and more attention. Fulfilling this task is no easy job since various types of food deteriorate differently and attract numerous food microbes. On that account, a specific formulation of edible functional coating/packaging is necessary to provide adequate protection for food. The vast number of published articles that evaluate the possible utilization of natural preservatives and ingredients for the production of edible coatings implies the necessity for dealing with this problem. However, at the moment only a small number of patents are available regarding this topic, and the fact that all of them are of newer dates indicates that this technology is still under development. In this field, sensory change due to the application of nano-structured edible coatings (especially those with essential oils) is an important issue since this can decrease consumer acceptance. Of the 16 presented patents, only two patent documents included the sensory study of coated fruits and vegetables. Therefore, future research should focus on enhancing the sensory qualities of edible coatings as well as presenting obtained results that are valuable to the scientific community. There is huge potential for the application of plant-based products, but to reach the point of its exploitation on an industrial level, it needs further improvements and optimization. Even though it is clear that more studies have to be conducted in order to reach the point when this technology can economically replace synthetic products, inevitably a lot of effort will be put towards this aim, making it both profitable and sustainable in the future.

## Figures and Tables

**Figure 1 foods-12-02535-f001:**
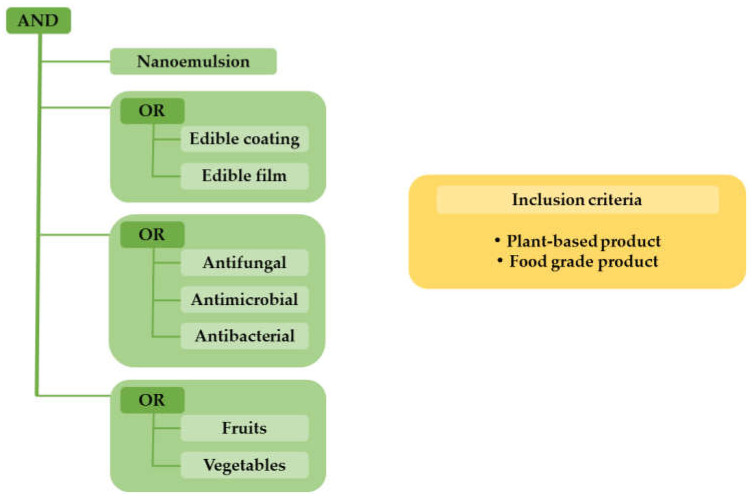
Patent search strategy composed of keywords and inclusion criteria.

**Figure 2 foods-12-02535-f002:**
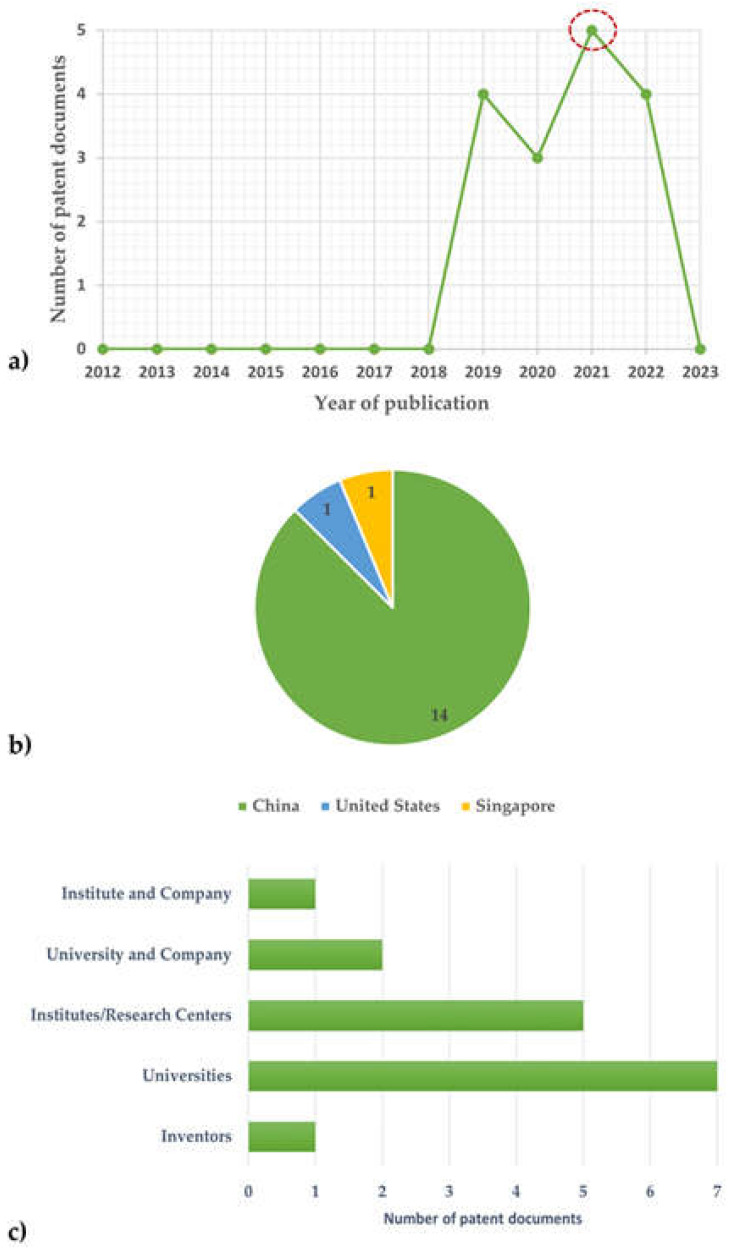
Publication dates of patents (**a**) and distribution of found patents by country (**b**), as well as the types of affiliation and partnerships involved in the patents (**c**) that concern plant-based nano-emulsions as edible coatings in the extension of fruit and vegetable shelf life. The search was filtered to include only the patent documents for the period from January 2012 to April 2023. In subfigure (**a**), a red circle points out that the highest number of patent publications, in total 5 patents, was in 2021.

**Table 1 foods-12-02535-t001:** List of patents that met the inclusion criteria. Data were extracted from patent records [17,18,19,20,21,22,23,24,25,26,27,28,29,30,31,32].

ID	Country	Patent Number	Patent Name	Publication Year	Inventor	Applicants	Summary
1	US	WO2020010173A1	Lipid nano-emulsion-doped antimicrobial packaging film	2020	Valentina Trinetta; Umut Yucel	Kansans State University	Food-grade emulsions with sub-micron droplets are used to encapsulate essential oils within a carrier in pullulan-based film packaging systems.
2	CN	CN202110462055A	Application method of zingiber corallinum hance essential oil in preparation of edible spray for prolonging shelf life of vegetables and fruits	2021	Zhao Tianming; Yang Haiyan; Zhang Zhen	Guizhou Institute Technology	This invented edible spray uses molecular distillation technology to enrich the antioxidant components in coral ginger essential oil and then prepares it into a nano-emulsion and mixes it with a film-forming liquid made of modified chitosan.
3	CN	CN112849496A	Modified atmosphere packaging method for fresh-cut fruits and vegetables by combining nano-coating film with microporous film	2021	Zhang Min; Wu Dan; Guo Zhimei	Wuxi Chemical Equipment Co., Ltd. Jiangnan University	The invention discloses a modified atmosphere packaging method for fresh-cut fruits and vegetables by combining a nano-coating film with a micro porous film.
4	CN	CN111278288A	Method for producing a nano-emulsion with encapsulated natural antioxidants for preserving fresh and minimally processed foods, and the nano-emulsion thus produced	2020	Malanati Ramos Miguel Enrique Jesus; Adriazola Du-Pont Melissa Ximena; Oviedo Morales Daniel Ali	Malanati Ramos Miguel Enrique Jesus	The invention relates to a method for producing a nano-emulsion comprising encapsulated natural antioxidants, disclosing the formulation of said product and the application thereof for the preservation of different natural food products or derivatives.
5	SG	WO2021107878A1	Edible coating films compatible with vegan, halal, and kosher diets for preservation of fruits and vegetables	2021	Maria Antipina; Daria Belova	Agency for Science, Technology and Research	Herein disclosed is a film-forming composition which includes a hydrogel comprised of a polysaccharide and a water-soluble additive including a phenolic phytochemical and an anti-fungal agent, wherein the hydrogel is formed as a matrix incorporated with the water-soluble additive.
6	CN	CN110236070A	Antibacterial emulsion with pure essential oil, preparation method, and application of emulsion	2019	Li Yan; Zhao Runan; Hu Junjie; Liu Shilin; Li Bin	Huazhong Agricultural University	The antibacterial emulsion is prepared from plant essential oil, flavor aldehyde, chitosan, a surfactant, etc. The antibacterial activity of the prepared emulsion is superior to that of single plant essential oil, the content of the surfactant is low, the emulsion does not contain edible fat and oil, and the preparation conditions are mild.
7	CN	CN110178892A	Essential oil nano-emulsion emulsified by xanthan gum and preparation method and application thereof	2019	Wang Dan; Zhang Xuedan; Zhang Quian; Yang Juanxia; Zou Man; Zhang Jing; Zhai Hao; Xin Li	Shandong Institute of Pomology	The invention discloses an essential oil (mainly clove essential oil) nano-emulsion emulsified by xanthan gum and starch octenyl succinate. This nano-emulsion particularly can solve the problems of easy rot and browning of fresh-cut apples in the prior art.
8	CN	CN110810091A	Fresh-keeping method for fruits and vegetables	2020	Zou Liqiang; Liu Wei; Xu Jong; Wang Pengze; Zhou Lei; Miao Jinyu; Zhou Wei; Liu Junping	Nanchang University	The embodiment of the present invention provides a method for fresh-keeping fruits and vegetables, which uses double emulsions to load cinnamon essential oil and salicylic acid.
9	CN	CN113207957A	Preparation method of rhizoma zingiberis recens essential oil nano-emulsion	2021	Chen Xiaoyu; Lou Zaixiang; Yu Jiayu; Wang Hongxin; Li Yaqin	Jiangnan University	The invention provides a method for preparing ginger essential oil nano-emulsion and the application of the prepared essential oil nano-emulsion in tomato preservation, which solves the problems of low water solubility, high volatility, etc., of plant essential oils.
10	CN	CN114271321A	Composite natural preservative as well as preparation method and application thereof	2022	Feng Qiqin; Zhong Fangije; Zhou Bingxian; Xu Qiongjun; Chen Hanpeng; Li Zhenxin	Hainan Medical College	The present invention takes Hainan cabbage as a research object and evaluates the effects of tea saponin, tea leaf essential oil nano-emulsion, and chitosan and selects the best ratio to formulate a green, safe, and effective compound natural preservative.
11	CN	CN109601612A	Cinnamon essential oil loaded Pickering emulsion and preparation method thereof	2019	Wang Dan; Jiang Yang; Zhang Jing; Zhang Qian; Zhai Hao; Zou Man; Xin Li	Shandong Institute of Pomology	The invention discloses cinnamon-essential-oil-loaded Pickering emulsion. The cinnamon-essential-oil-loaded Pickering emulsion is characterized by comprising the following components: compounded oil phase where cinnamon essential oil is a main component, zein solution, pectin solution, and xanthan gum solution.
12	CN	CN115399414A	Fruit nano-coating preservative formula and production method	2022	Sun Shunguang; Lan Wenzhong; Ji Li; Cui Minghao; Fang Tianhe; Zhang Xinming	Shandong Food Fermentation Industry Research and Design Institute	The invention provides a fruit nano-coating preservative formula that comprises the following raw materials: isobutyl paraben, ethylene oxide higher fatty alcohol, octenyl succinic acid curdlan gum ester, surfactant, and water. The stable nano-emulsion takes water as a dispersion medium, is uniform in film coating, and is safe and reliable.
13	CN	CN114698688A	Preparation and application of nano-structure lipid carrier based on medium-long-chain diglyceride-embedded litsea cubeba essential oil	2022	Qiu Chaoying; Yu Yasi; Wang Yong; Li Ying; Zhang Zhen; He Jiajing	University of Jinan Qingyuan Yaokang Biotechnology Co., Ltd.	The invention explains a method for the production of a nano-structure lipid carrier based on medium- and long-chain diglyceride-embedded *Litsea cubeba* essential oil, while manifesting a small particle size, high encapsulation efficiency, and good stability.
14	CN	CN110063366A	Preparation method of lacquer wax composite nanometer silver liquid antibacterial coating preservative	2019	Wang Chengzhang; Xue Xingying	Nanjing Tianbang Biological Technology Co., Ltd. Institute of Chemical Industry of forest Product CAF	The invention discloses the method for the formulation of a lacquer wax composite nanometer silver liquid antibacterial coating preservative unloaded with ginkgo leaf lipid compound. It has an antibacterial effect on *Staphylococcus aureus*, *Bacillus subtilis*, and other fungi and has increased efficiency compared to a single coating of lac.
15	CN	CN114651868A	Black pepper essential oil nano-emulsion as well as preparation method and application thereof	2022	Pan Yonggui; Nie Yudong	Hainan University	The invention proposes a black pepper essential oil nano-emulsion with good antibacterial activity, obtained from black pepper essential oil, selected surfactants and co-surfactants, and water.
16	CN	CN113712072A	Clove essential oil Pickering emulsion coating preservative and preparation method and use thereof	2021	Xu Baocai; Hua Lu; Deng Jieying; Wang Zhaoming; Zhou Hui	Hefei University of Technology	The invention is related to a clove essential oil Pickering emulsion coating preservative, in which the following formulation is included: clove essential oil, zein powder, sodium caseinate, and chitosan.

## Data Availability

Data are contained within the article. The data used to support the findings of this study can be made available by the corresponding author upon request.

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
