# Peer review of "Plant-Based Nano-Emulsions as Edible Coatings in the Extension of Fruits and Vegetables Shelf Life: A Patent Review"

_foods, 2023, doi:10.3390/foods12132535_

Round 1

Reviewer 1 Report

This is a patent review in the field of plant-based nanoemulsions used as edible coatings to extend fruits and vegetables shelf-life. Based on the patents reviewed in this field only 16 patents are available and 14 of them are from China. The field is very specific and not enough for a good review article for a Q1 journal. 

Also, something may not be informative, your search year range is from 2012 to April 2023. In 2023 you can not report it as Zero!

So, I recommend expanding the field of search to at least 100+ patents or modifying this as a mini review and submit in other journals.

Reviewer 2 Report

The Authors provided a scientific scan for the current patents which might be applied in food preservation. The manuscript content is generally well-organized and suitable for the journal's aims.

My recommendation is to accept after minor improvements. Detailed comments were listed below:

Abstract

- consider changing the first part - not mandatory. In my opinion, might be more focused on the coatings.

Introduction

- this part is well organized

- it should be pointed out that references to previous reviews as well as short historical mentions are interesting.

Section 2

- line 86 - please add some examples of the preservation which was used

Section 3

- is well organized and clearly presents the search strategy

Section 4

- figure 2 please improve the quality, Furthermore please have a look for a left down corner there is some unnecessary bracket )

- if possible some graphs/images/figures can be applied - not mandatory, but it will increase the manuscript content quality

Reviewer 3 Report

Dear author,

Many thanks for your good review. My comments are in the attached file.

Best regards

Round 2

Reviewer 1 Report

I am satisfied with the justification.

Reviewer 3 Report

Dear author,

Thanks for the revised file.

Regards

The quality of English is good. However, please check again for grammatical errors.

Regards